# Circulating Phylotypes of White Spot Syndrome Virus in Bangladesh and Their Virulence

**DOI:** 10.3390/microorganisms10010191

**Published:** 2022-01-16

**Authors:** Mehedi Mahmudul Hasan, M. Nazmul Hoque, Firoz Ahmed, Md. Inja-Mamun Haque, Munawar Sultana, M. Anwar Hossain

**Affiliations:** 1Department of Microbiology, University of Dhaka, Dhaka 1000, Bangladesh; mehedimahmudul@gmail.com; 2Department of Fisheries and Marine Science, Noakhali Science and Technology University, Noakhali 3814, Bangladesh; 3Department of Gynecology, Obstetrics and Reproductive Health, Bangabandhu Sheikh Mujibur Rahman Agricultural University, Gazipur 1706, Bangladesh; nazmul90@bsmrau.edu.bd; 4Department of Microbiology, Noakhali Science and Technology University, Noakhali 3814, Bangladesh; firoz19701016@gmail.com; 5Department of Fisheries, University of Dhaka, Dhaka 1000, Bangladesh; injamamunhaque@du.ac.bd; 6Vice-Chancellor, Jashore University of Science and Technology, Jashore 7408, Bangladesh

**Keywords:** VP28, WSSV, phylotypes, real-time PCR, viral load

## Abstract

White Spot Syndrome Virus (WSSV) has emerged as one of the most prevalent and lethal viruses globally and infects both shrimps and crabs in the aquatic environment. This study aimed to investigate the occurrence of WSSV in different ghers of Bangladesh and the virulence of the circulating phylotypes. We collected 360 shrimp (*Penaeus monodon*) and 120 crab (*Scylla* sp.) samples from the south-east (Cox’s Bazar) and south-west (Satkhira) coastal regions of Bangladesh. The VP28 gene-specific PCR assays and sequencing revealed statistically significant (*p* < 0.05, Kruskal–Wallis test) differences in the prevalence of WSSV in shrimps and crabs between the study areas (Cox’s Bazar and Satkhira) and over the study periods (2017–2019). The mean Log load of WSSV varied from 8.40 (Cox’s Bazar) to 10.48 (Satkhira) per gram of tissue. The mean values for salinity, dissolved oxygen, temperature and pH were 14.71 ± 0.76 ppt, 3.7 ± 0.1 ppm, 34.11 ± 0.38 °C and 8.23 ± 0.38, respectively, in the WSSV-positive ghers. The VP28 gene-based phylogenetic analysis showed an amino-acid substitution (E→G) at the 167th position in the isolates from Cox’s Bazar (referred to as phylotype BD2) compared to the globally circulating one (BD1). Shrimp PL artificially challenged with BD1 and BD2 phylotypes with filtrates of tissue containing 0.423 × 10^9^ copies of WSSV per mL resulted in a median LT50 value of 73 h and 75 h, respectively. The in vivo trial showed higher mean Log WSSV copies (6.47 ± 2.07 per mg tissue) in BD1-challenged shrimp PL compared to BD2 (4.75 ± 0.35 per mg tissue). Crabs infected with BD1 and BD2 showed 100% mortality within 48 h and 62 h of challenge, respectively, with mean Log WSSV copies of 12.06 ± 0.48 and 9.95 ± 0.37 per gram tissue, respectively. Moreover, shrimp antimicrobial peptides (AMPs), penaeidin and lysozyme expression were lower in the BD1-challenged group compared to BD2 challenged shrimps. These results collectively demonstrated that relative virulence properties of WSSV based on mortality rate, viral load and expression of host immune genes in artificially infected shrimp PL could be affected by single aa substitution in VP28.

## 1. Introduction

Shrimp aquaculture is one of the major earning sources in many countries including Bangladesh and plays a vital role in enlightening community advancement, food security, employment opportunities and poverty reduction [1,2]. In Bangladesh, shrimp aquaculture provides livelihood to around 85 million people (mostly coastal people) and serves as the second most foreign currency-earning source, which contributes about 5% to national GDP [3,4]. Black tiger shrimp (*Penaeus monodon*) contributes 26% to the total aquacultural production in Bangladesh, while crabs contribute 6% [5]. Studies suggested that *P. monodon* can adapt to a wide range of salinity from 4 to 40 ppt, dissolved oxygen from 4 to 7 ppm, temperature from 25 °C to 32 °C, and pH from 7.5 to 9 [6,7]. The mud crab, *Scylla* spp., is distributed widely throughout the Indo-Pacific region [8]. Recently, mud crab (*Scylla olivacea*) farming has had an increasing trend in the coastal areas of Bangladesh due to their higher disease resistance capacity and market values [9]. Among the marine crustaceans, crabs are supposed to be less vulnerable to the effects of climate change and deterioration of water quality. To date, more than 98 species have been found as hosts or carriers of WSSV [10], and of them, mud crabs have been considered to be a particularly potential threat to shrimp farms because of their carrier status [10,11,12]. Moreover, mud crabs may well suppress viral replication by inducing the apoptosis of hemocytes [13].

The WSSV is one of the major threats to the shrimp industry over the past two decades globally. This is a very fast reproducing, wide spreading and highly virulent crustacean pathogen [14,15]. Studies have demonstrated the widespread pathogenicity of WSSV among many marine crustaceans, including shrimp, crayfish and crabs [13,16]. The outbreak of WSSV depends on the interactions among the pathogen, host and environment [17], and thus, the interaction between WSSV and the hosts have been a research focus in recent years. This virus can be transmitted both horizontally and vertically [12,18], and once an outbreak of WSSV occurs, it wipes out the entire population in many aquatic farms within a few days [14]. Infection of the WSSV in shrimp is characterized by a rapid mortality of up to 100% within 7–10 days [19]. Up to now, there is no completely efficient method to protect the shrimp from WSSV infection. Appropriate protective management is enormously important for reducing WSSV infections in shrimp farms [20]. In Bangladesh, eggs are hatched in the hatcheries from mother shrimps collected from the Bay of Bengal, and shrimp post larvae (PL) from these hatcheries are distributed throughout the country. PL traders directly catch and sell to shrimp farmers as well. Breeding of the WSSV-resistant types should be the most effective approach towards solving the virus disease problem [15]. However, comprehensive study regarding the physicochemical parameters of ghers (shrimp ponds) of Cox’s Bazar and Satkhira districts of Bangladesh, and identification of circulating WSSV in those areas are lacking.

The circular genome of the WSSV is approximately 275 nm in length and 120 nm in width with tail-like appendages at one end, and composed of five known major structural proteins: VP28, VP19, VP26, VP24 and VP15 [19,21]. Studies on WSSV viral proteins have demonstrated that VP28 and VP19 are associated with the virion envelope [21,22], while VP26 acts as a tegument protein linking the two nucleocapsid-associated proteins VP24 and VP15 to the envelope [17,23]. The VP28 is required for WSSV entry into host cells by endocytosis, cell-to-cell infection and virus propagation [22]. Moreover, this envelop protein plays a significant role in initiating the WSSV infection in shrimp [22]. The VP28 gene of WSSV is suggestively involved in endosome escape through its interaction with the host Rab7 [24], and has been identified as a potent target for dsRNA treatment in comparative studies [25]. Previous analyses of strain variability reported that competitive fitness depends on the size of the genome [26,27]; however, these studies are far from enough to illustrate all the mechanisms of WSSV pathogenesis. The on-going mutations in the structural proteins such as VP28 could explain most phenotypic variance of WSSV for certain traits [15,28]. Siddique and colleagues confirmed a mutation at amino acid position 167 of the VP28 gene of WSSV in Bangladesh with glycine instead of glutamic acid on that position [29]. However, the virulence properties of different phylotypes of WSSV remain unknown. Therefore, this study investigated the impact of physicochemical parameters on the prevalence of WSSV, the virulence properties of circulating phylotypes of WSSV based on mortality observation and the viral load count in shrimp PL artificially infected with two circulating phylotypes of WSSV.

## 2. Materials and Methods

### 2.1. Sampling and Measurement of Physicochemical Parameters

Shrimp (*P. monodon*) and crab (*Scylla* sp.) samples were collected from Sadar Upazilla of Cox’s Bazar District and 5 Upazillas namely: Satkhira Sadar (SS), Debhata (D), Asassuni (A), Kaliganj (K) and Shyamnagar (S) of the Satkhira District which are situated in the south-east and south-west coastal regions of Bangladesh (Figure 1). Satkhira is situated in the southernmost coastal region of Bangladesh, and approximately 32% of the area of this district (nearly 66,800 hectors out of 211,000 hectors) consists of shrimp farms and contributes 34% (23,400 metric ton out of 69,000) of the entire shrimp production. Cox’s Bazar is situated in the south-east coastal region of Bangladesh where most of the shrimp hatcheries are situated and approximately 41,594 ha is under shrimp and crab aquaculture [5]. In this study, samples were collected from 20 ghers during the period when farmers reported the presence of WSD in the area (five from Cox’s Bazar and fifteen from Satkhira). A total of 360 shrimps and 120 crabs were grossly collected after farmers complained of the death of the crustaceans in their farming ghers (shrimp ponds) during monsoon season (May-June) in 2017 to 2019. We also continued our sampling in post-monsoon season (October) during the three-year study period (2017–2019) when no death of crustaceans reported in the study ghers by the respective farmers. Data on temperature, pH, dissolved oxygen and salinity were collected from the study ghers.

### 2.2. DNA Extraction

After initial screening considering symptoms of disease, tissue DNA was extracted from the collected samples (from both shrimp and crabs) by an automated DNA extraction system (MaxWell 16^®^ Tissue DNA Purification kit; AS 1030, Promega, Madison, WI, USA), according to manufacturer’s instruction [29]. In addition, DNA was also extracted from artificially challenged shrimp PL tissues. For this, challenged shrimp PL tissue were collected in sterile 1.5 mL microfuge tubes and mashed into fine particles with a glass rod prior to DNA extraction. DNA concentration and purity were measured by Nano-Drop 2000 (Thermo Scientific, Waltham, MA, USA) [20].

### 2.3. Conventional and Quantitative Real-Time PCR (qPCR) Assay

The extracted DNA underwent conventional PCR assay for the amplification of VP28 gene using GoTaq 2 ×Hot Start Colorless Master Mix (Promega, Madison, WI, USA) with forward and reverse primers [29,30]. The conventional PCR reactions included denaturation at 95 °C for 50 min, annealing at 55 °C for 30 s, extension at 72 °C for 45 s, and repeated for 30 cycles with a final extension of 5 min at 72 °C. We used 1.0% agarose gel (with ethidium bromide staining) in TAE buffer to separate and visualize the PCR-amplified products. Following electrophoresis, the bands were photographed under UV light (Appendix A).

The real-time qPCR was performed with primer pair WSSV-q28F 5′-TGTGACCAAGACCATCGAAA-3′ and WSSV-q28R 5′-CTTGATTTTGCCCAAGGTGT-3′ following previously developed methods [29] with a few adjustments. In the current study, recombinant plasmid-based standard was used instead of the purified PCR product-based standard [29,31]. The recombinant plasmid that contained the VP28 gene (TOPO TA Vector with complete CDS of VP28 gene as an insert) was gel purified by using the Wizard^®^ SV Gel and PCR Clean-Up System (Promega, Madison, WI, USA). From a serial dilution of the purified recombinant plasmid, the standard was prepared in a linear logarithmic scale of 1.0 × 10^9^ to 10^2^ copies per reaction. In brief, all the qPCR reactions had a final volume of 25 µL and were run in the Applied Biosystems^®^ 7500 Real-Time PCR system (Foster City, CA, USA) by using 2 × SYBR^®^ Green PCR Master Mix (Applied Biosystems, Foster City, CA, USA), 100 nM of the forward and reverse primers and variable quantity of each template DNA. The parameters for thermal cycling were set for an initial denaturation step at 95 °C for 10 min followed by 40 cycles at 95 °C for 15 s for DNA denaturation with subsequent annealing and extension at 53 °C for 30 s. The melt curve was analyzed to differentiate the specific amplicon formed by primer dimer or generation of any non-specific product. Furthermore, the qPCR products were electrophoresed in agarose gel to invalidate the existence of any spurious amplicon. The experiment was performed with duplicate replication for the purpose of quantifying the viral load. The WSSV load per gram of tissue sample was calculated using the following equation:Viral load per gram tissue = [viral load per reaction × (Final Elution volume/volume of template DNA per reaction) × dilution factor] ± Standard Deviation (SD).

To evaluate the standard curve’s reproducibility, standard reactions were performed three times separately, including duplications of each reaction. The data obtained from real-time PCR run were analysed using 7500 software, version 2.0.6 (Applied Biosystems, Foster City, CA, USA). Statistical program Microsoft Excel 2020 was used to analyze the data, which were presented as mean ± SD. The standard deviation of the viral load per reaction was considered during the viral load calculation. For relative virulence study, viral copy numbers in the challenged shrimp PL were quantified per reaction with the same amount of the initial concentration of DNA. The VP28 gene amplified through real-time PCR produced a 148 bp product (Appendix A).

### 2.4. Sequencing of VP28 Protein, Phylogenetic and Mutation Analyses

Conventional PCR-amplified products were purified with the Wizard^®^ SV Gel and PCR Clean-Up System (Promega, Madison, WI, USA), and the seven (shrimp = 5; crab = 2) purified PCR products were exposed to an automated dideoxy cycle sequencing reaction using BigDye^®^ Terminator v3.1 cycle sequencing kit (Applied Biosystems^®^, Foster City, CA, USA) according to manufacturer’s instruction [20]. A sequence cleaner (https://github.com/metageni/Sequence-Cleaner, accessed on 20 June 2020) with set parameters of minimum length (m = 3822), percentage N (mn = 0), keep_all_duplicates, and remove_ambiguous was used to remove all ambiguous and low-quality sequences [32]. The raw sequence data were assembled through SeqMan version 7.0 (DNASTAR, Inc., Madison, WI, USA) and the assembled sequences were compared with other entries from NCBI GenBank [33] with BLAST [34] search to disclose the identification and matching with VP28 gene of WSSV.

Using Molecular Evolutionary Genetics Analysis (MEGA) version 7.0 for the larger datasets [35], the VP28 gene sequences, amplified from seven isolates, were aligned with each other, and with relevant reference sequences from our previous study (*n* = 17) and NCBI GenBank database (*n* = 10), with >90% taxonomic identity. A maximum-likelihood tree was generated with the Tamura-Nei evolutionary model [35,36]. Nodal confidence in resulting phylogenetic relationships was evaluated using bootstrap test (1000 replicates) [37]. Seven VP28 sequences of WSSV different isolates of shrimp and crab and 143 reference sequences of VP28 retrieved from GenBank were subjected to multiple alignment through MAFFT [38], and some adjustments were made by manual editing. Repeat units of each isolate were annotated using Geneious Prime (Trial Version), and aligned against a reference sequence for amino-acid (aa) variability score counting [39,40].

### 2.5. Experimental Infection

*P. monodon* post larvae (PL) were collected from Meghna Shrimp Hatchery, Cox’s Bazar, Bangladesh with a length of 1–2 cm. WSSV-free breeder wild-caught mother shrimps’ eggs were used for hatching these post larvae in the hatchery. Highly infected black tiger shrimp tissue was used for preparing WSSV inoculum. Tissue below carapace from affected shrimp was minced and homogenized in sterile sea water. Supernatant was collected after centrifugation at 8515× *g* for five min and filtered through a 0.45 µm membrane. Stock was diluted to prepare infective dose containing 10^8^ copies per mL. Blank inoculums were prepared using the same steps from a WSSV-negative shrimp sample. For in vivo challenge, we performed the previously established ‘immersion technique’ [41] as waterborne inoculation [42]. The experimental groups were challenged by the immersion technique in aerated glass jars (*n* = 180, in each jar) with WSSV solutions in three treatments. Using the WSSV-negative inoculums, aerated jars were set to treat the negative control group (*n* = 180 in each jar) with three treatments. The PL were fed with commercially available artificial feed once a day at a rate of 10% body weight. Mortalities were checked after every six hours, and the presence of WSSV was checked by conventional PCR (Appendix A).

Virulence determination assay was performed in separate experiments with three treatments of shrimp PL (*n* = 360, in each jar) infected with inoculums containing the same copy number of both phylogroups (BD1 and BD2). A negative control group with three treatments (*n* = 360, in each jar) was maintained with inoculum prepared from the WSSV-negative tissue. So, the total number of jars used was nine. Thus, inocula were prepared using the abovementioned procedure from tissue with same copy numbers of the virus (4.27 × 10^9^) from both groups and added in the small aquariums for treatments. Infective doses consisted of filtrates from the tissue of both groups containing 0.423 × 10^9^ and 0.423 × 10^7^ copies of WSSV per mL in sterile sea water. The temperature of the experimental tanks’ water ranged between 28–29 °C, while the salinity and dissolved oxygen were 18 ppt and 5–6 ppm, respectively, throughout the experiment. Time-dependent mortality rates were measured by counting the number of dead PL in every six-hour interval. DNA from challenged PL (infected and control) were extracted and tested for viral load estimation by real-time PCR. C_T_ values and copy number of WSSV obtained from different samples were compared.

Another pilot study of infection assay was conducted on 45 mud crabs using the tissues from both phylotypes (BD1 and BD2) through the ingestion method. The crabs were collected from a certified WSSV-free crab farm of Satkhira district. Moreover, we also tested the health status of the crabs through conventional PCR before experimental infection. The crabs were divided into three groups: Group-1 = 15 crabs challenged with BD1, Group-II = 15 crabs challenged with BD2, and Group-III = 15 crabs, serving as control. The crabs were of 6 months of age, and each group was kept in a separate tank containing 5 crabs in each. Crabs of the treated groups were fed with infected crab tissue (Day 1) and trash fish (from Day 2), while crabs of the control group were given beef liver from the first day [42]. Tissues containing both groups of WSSV (1.0 × 10^9^) were used to feed the crabs of the experimental groups. Standard experimental conditions were similar to the shrimp PL virulence experiment other than the salinity, which was maintained at a higher level (28–30 ppt).

### 2.6. RNA Extraction and Preparation of cDNA

RNA was extracted from the challenged PL at the moribund stage using QIAgen’s QIAamp Viral RNA mini kit according to the protocol of manufacturer. The purity and concentration of the extracted RNA were assessed by Nanodrop-2000 spectrophotometer. The extracted RNA was reverse-transcribed to cDNA using New England Biolab’s cDNA kit (PhotoScript II First Strand cDNA Synthesis Kit, New England Biolabs, Ipswich, MA, USA).

### 2.7. Gene Expression Analysis

Real-time PCR was used for the analysis of expression of two immunity genes of shrimps (Penaeidin and Lysozyme). Quantitative PCR was run with one cycle of initial denaturation at 95 °C for 60 s and 45 cycles of denaturation at 95 °C for 15 s and extension at 60 °C for 30 s using New England Biolab’s Luna Universal qPCR master mix according to the protocol of manufacturer. The primers used are presented in Table 1. Average-fold difference in gene expression was analysed by comparative delta C_T_ method [43]. Housekeeping gene *beta-actin* was used for normalization that provided the C_T_ values as internal (endogenous) control. Control treatments were inoculated with inoculum prepared from WSSV-negative tissue, and the samples were considered for gene expression if there was any amplification. Delta C_T_ has been calculated by deducting C_T_ value of endogenous control from the gene of target, and finally, mean delta C_T_ was calculated from this standardized delta C_T_ value. Delta C_T_ was calculated with reference to the control by deducting the mean delta C_T_ of the control from the mean delta C_T_ of the target gene. Changes in average-fold gene expression in challenged PL was calculated to 2^-delta delta C^_T_ values.

### 2.8. Observation of Binding Affinity of VP28 and Its Receptor Protein Rab7

The web server PRODIGY was used to observe the binding affinity of the receptor protein Rab7 and the WSSV envelope protein VP28 of both groups from Bangladesh [46,47]. VP28 with glycine at its 167th amino acid position and VP28 with glutamic acid at 167th amino acid position were used for docking with Rab7 of *P. monodon* by HADDOCK (High Ambiguity Driven protein–protein DOCKing) [48]. For protein–protein docking, active residues [48] were produced from both molecules. Prior to docking, the Rab7 sequences were downloaded from the NCBI database and the homology modelling of Rab7 protein was conducted using SWISS-MODEL and validated further by developing Ramachandran plots [49]. PROGIDY used interactors from both proteins for producing binding affinity (∆G) and the dissociation constant (Kd) values. Moreover, DynaMut web server was used to observe the impact of this mutation to the stability of the VP28 [50].

### 2.9. Statistical Analysis

The salinity, temperature, dissolved oxygen and pH were compared to WSSV, grouped as present and absent. In the case of salinity, temperature and dissolved oxygen, non-parametric independent sample tests were performed, and for pH, a parametric independent sample test was performed. The null hypothesis of the distribution of physicochemical parameters was the same across categories of WSSV and was retained for dissolved oxygen and pH and rejected for salinity and temperature. When there is no significant interaction between time and isolate, the probit model takes the form: probit (x) = α + β time + γ isolate, where α is the intercept, β is the rate of probability change per unit change of time (for a constant isolate), and γ is the rate of probability. Eta test statistic was used to study the associations between WSSV (nominal variable) and the physicochemical parameters (scale variables). Additionally, the mean prevalence of WSSV among different regions was calculated using SPSS (SPSS, Version 23.0, IBM Corp., New York, NY, USA) [51]. Final values for average-fold differences in gene expression were tested for significance at the 5% level using *t*-test.

## 3. Results

### 3.1. Prevalence of WSSV and Physicochemical Parameters in the Study Ghers

The overall prevalence of WSSV in both crustacean population (shrimp and crab) between the study areas (Cox’s Bazar and Satkhira) and over the study periods (2017–2019) differed significantly (*p* < 0.05, Kruskal–Wallis test). The average prevalence of WSSV during 2017 to 2019 in shrimp and crabs were 20.93% and 12.73% in the ghers of Cox’s Bazar, and 16.73% and 9.53% in the ghers of Satkhira, respectively (Figure 2A). By comparing the year-wise prevalence of WSSV in the shrimp between the study areas, the highest prevalence was recorded in the shrimp ghers of both Cox’s Bazar (23.11%) and Satkhira (18.96%) in 2017, and WSSV infection rates in shrimp population gradually decreased thereafter in 2018 and 2019 (Figure 2A). Conversely, the prevalence of WSSV in the crab population was found to be highest in the ghers of Cox’s Bazar (14.16%) in 2019 and Satkhira (10.54%) in 2018 (Figure 2A). Physicochemical parameter analysis of the study ghers showed that the mean value of salinity, dissolved oxygen, temperature and pH ranged from 14–16 ppt, 3.6 to 3.8 ppm, 33.9 to 34 °C and 7.8–8.4, respectively, in the WSSV-positive shrimp ghers, and 13–21 ppt, 2.61 to 6.06 ppm, 28.7 to 34 °C and 8.2 to 9.1 in WSSV-negative ghers (Table 2 and Appendix A). Comparing the physicochemical parameters across the study ghers, we found higher water temperature in WSSV-positive ghers compared to WSSV-negative ghers. Likewise, the WSSV-negative ghers also had higher salinity than WSSV-positive ghers (Figure 2B).

### 3.2. Detection of WSSV in Scylla Olivacea

One of the hallmark findings of this study was the detection of WSSV in the mud crabs (*Scylla olivacea*) in the samples of both regions, Cox’s Bazar and Satkhira. The appearance of a 643 bp PCR product confirmed the presence of WSSV in the crabs (Appendix A).

### 3.3. Viral Loads in Circulating Phylotypes of WSSV Differed in Crustacean Samples

In the current study, WSSV Log load per gram of shrimp tissue ranged from 7.62 (Cox’s Bazar) to 12.35 (Satkhira), while that of crab tissue ranged from 8.20 (Cox’s Bazar) to 10.47 (Satkhira) (Figure 3). On average, shrimp samples from Satkhira had significantly higher (*p* < 0.05, Kruskal–Wallis test) viral load (10.48 ± 0.32, SEM) than that of Cox’s Bazar (8.40 ± 0.08, SEM). The viral load in crab samples also varied significantly (*p* < 0.05, Kruskal–Wallis test) in the study areas, keeping a higher load (9.92 ± 0.56, SEM) in Satkhira compared to samples from Cox’s Bazar (8.91 ± 0.72, SEM) (Figure 3).

The VP28 gene-based phylogenetic analysis divided the isolates into two major phylotypes (BD1 and BD2) which are currently circulating across the country. Nucleotide sequences obtained from seven crustacean isolates according to VP28 gene sequencing (2017–2019) along with 17 previously reported reference sequences (2014–2015) from Bangladesh (our laboratory) and 10 reference sequences retrieved from NCBI database were used to generate a phylogenetic tree. Two clusters (BD1 and BD2) contained 24 of the sequences isolated from Bangladesh that mostly related to isolates from India and Vietnam. The phylotype BD1 and BD2 contained 8 and 16 isolates of WSSV, respectively (Figure 4A). Two isolates (MZ383193 and MZ383194) belonging to the phylotype BD1 had 98–100.0% similarity with six VP28 gene sequences reported previously from Bangladesh (2015 and 2017), and that of two Indian sequences (Figure 4A). The isolates of the BD1 phylotype formed a slightly distant branch with VP28 gene sequences reported from other countries such as China, South Korea and Vietnam. On the other hand, the isolates of BD2 phylotype (MZ383195-198) were found to be closely clustered with other VP28 sequences that was previously reported (2014–2015) to be circulated in Bangladesh (Figure 4A).

### 3.4. Variations in Amino-Acid Mutations in VP28 of WSSV

For amino acid (aa) mutation analysis, we used one VP28 reference sequence from the Thai isolate (GenBank Accession no. AF369029). Out of 150 VP28 sequences retrieved from the NCBI database (including 24 sequences from our laboratory), 104 (69.33%) sequences showed aa mutations at 21 positions. Among these mutations, residue position 42 (33.33%), 114 (8%) and 167 (12.67%) were found as the major mutation sites in the VP28 sequences of WSSV (Appendix A). However, residue position 167 showed glycine instead of glutamic acid (E→G) in 19 VP28 sequences, including 15 from Bangladeshi and four from Egyptian sequences (Appendix A). Of the Bangladeshi sequences, 10 were obtained from previous studies from our laboratory [20,29], and the rest of the five (GenBank Accession no. MZ383195-MZ383199) were from the current study. Moreover, in the current study, two VP28 sequences of both shrimp (GenBank Accession no. MZ383193) and crab (GenBank Accession no. MZ383194) isolates from Satkhira had E at the 167 residue position unlike the five sequences (GenBank Accession no. MZ383199; crab sample, and GenBank Accession no. MZ383195-MZ383198; shrimp samples) of Cox’s Bazar with G in that position (Figure 4B).

### 3.5. Shrimp Post-Larvae Mortality Rates and Lethal Time Differed between Phylotypes

The survival of shrimp post larvae (PL) between control and infected groups was statistically significant (*p* < 0.05, Kruskal–Wallis test) in all of our experiments of the current study. Shrimp PL when challenged with confirmed WSSV irrespective of the mentioned phylotypes showed a mortality rate of 96.67% at 96 h with a dose of 10^8^ copies of WSSV per mL of sterile seawater (Figure 5) in which the mortality observation took place every 6 h. In the present infection assay with three treatments, shrimps started to die at 48 h and lasted till 108 h, while shrimps in the control group did not die before those were sacrificed. By comparing the virulence assay between challenged doses, we found that shrimp PL challenged with Dose 1 (0.423 × 10^9^ copies of WSSV per mL) started to die after 66 h of challenge, and 100% mortality of PL occurred at 102 h after challenge with both phylotypes. However, using Dose 2 (0.423 × 10^7^ copies of WSSV per mL), the onset of shrimp PL mortality started at 72 h of challenge, and 100% mortality was found after 108 h and 114 h of challenge with BD1 and BD2, respectively (Figure 6 and Appendix A). The average median LT50 values were observed to be 73 and 75 h post infection (hpi), respectively, in BD1- and BD2-challenged shrimp PL with Dose 1, and 82 and 84 hpi with Dose 2 (Figure 6). However, this difference in median LT50 between the two phylotypes with both doses was not statistically significant (*p* > 0.05). In addition to these factors, doses and viral phylotypes, the shrimp PL’s physiological conditions and capacity to adapt in aquariums could also be the factors engaged in mortality. For that reason, viral load count in the infected PL tissue might help in revealing the study goal more accurately. Additionally, from that perspective, the quantification of the virus using real-time PCR assay was a major part of this study.

### 3.6. Quantitative Detection of WSSV in Challenged Shrimp PL

The mean Log viral copy numbers were 6.47 and 4.75 per mg of tissue, respectively in BD1- and BD2-treated PL, which was statistically significant (*p* < 0.05, Kruskal–Wallis test) with Dose 1 (Figure 7). In this study, we found an average C_T_ value of 20.01 and 25.32 in BD1- and BD2-challenged PL, respectively (Table 3, Appendix A). Figure 8 shows the positive amplifications in samples challenged with both phylogroups, positive controls and standards. In this quantitative assay, an extensive range of the mean viral Log load in challenged shrimp PL was observed, and the results showed that amplification curves were specific for samples, standards, negative controls and positive controls. No noteworthy fluorescence signal was noted for the negative control (NTC). C_T_ values for NTC were outside determination index, and C_T_ values in all positive samples had a quantitative index in qPCR, ranging from 7.19 to 33.48 (Figure 8A). Standard qPCR using the qVP28F and qVP28R primers revealed a single amplicon of 148 bp after agarose gel electrophoresis, indicating a specific amplification product. The standard curve generated was linear from log starting quantity of 2 to 9. The mean upper and lower quantification limits in challenged shrimp PL were 1.10 × 10^10^ and 3.31 × 10^2^ WSSV copies per mg tissue, respectively (Figure 8A,B).

### 3.7. Crab Mortality Rates and Viral Load Counts Differed in Treated Crabs

The LT50 and LT100 for infected crabs varied among the treatments, while no crab died in the control group. The experiment was run till 62 days post infection (dpi) as long as all crabs died in the experimental groups challenged with both phylotypes (BD1 and BD2) (Figure 9 and Appendix A). The LT50 and LT100 were 31 and 48 dpi in BD1-challenged crabs, and LT50 and LT100 were 40 and 62 dpi for BD2-challenged crabs, respectively. The mean viral loads in the crabs challenged with BD1 and BD2 were 12.06 ± 0.48 and 9.95 ± 0.37 per g of tissue, respectively (Figure 10, Appendix A).

### 3.8. Gene Expression Profiling of Immunity Genes in Both the Infected Groups

In the current study, two important immunity genes of shrimps, penaeidin and lysozyme, were considered for expression analysis, both of which are antimicrobial peptides (AMPs). Gene expression was observed after 73 hpi in this immersion challenge study. Figure 11 provides the average relative expression of penaeidin and lysozyme after exposure of shrimp PL to both the circulating phylotypes showing comparatively higher expression of lysozyme than penaeidin in both groups. While comparing the expression in the challenged groups, it was found that average relative expressions of these two genes were lower in BD1-challenged PL than in BD2-challenged PL (Figure 11).

### 3.9. Rab7-VP28 Binding Affinity

In the present study, the lower Kd value was observed for the binding of VP28 with glutamic acid at the 167th position (1.6 × 10^−8^) than the other mutated one with glycine on that position (5.1 × 10^−8^). The higher the Kd value, the lower may be the strength of binding between proteins (Appendix A). From this in silico approach towards getting the binding affinity in two different complexes, it was predicted that the BD1 might have more chance to bind with the receptor protein Rab7 of *P. monodon*. Moreover, prediction results from DynaMut showed an increase in molecular flexibility instead of rigidification in VP28 of BD2 through analysing the difference in vibrational entropy (ΔΔSVib ENCoM: 0.030 kcal·mol^−1^·K^−1^) (Appendix A).

## 4. Discussion

*Penaeus monodon* is considered as one of the most valuable commercially cultured aquatic species in Bangladesh. This crustacean species has been badly affected by WSSV in all shrimp-producing countries across Asia, including Bangladesh [20,52]. In this study, the prevalence of WSSV infections in crustacean populations (shrimps and crabs) varied in study areas, keeping significantly higher prevalence of this disease both in shrimp ghers of the Cox’s Bazar district compared to Satkhira. Compared to the shrimp population, the WSSV detection rate in crabs remained much lower, which might be associated with their disease tolerance capacity and carrier status [9]. Unlike shrimps, mud crabs are generally believed to be highly tolerant to WSSV and keep being infected for longer periods without symptoms of disease. In this study, the WSSV-positive crabs were found in those ghers only where the shrimps were also WSSV-positive. Rouf et al. reported in their study that the mud crab species found in the coastal regions of Bangladesh was *Scylla olivacea* [53]. They confirmed it by genetic analysis of the partial sequences of one mitochondrial gene, 12S rRNA, as well as through studying their morphological characteristics and morphometric ratios. However, if *P. monodon* and *Scylla* spp. are co-cultured, shrimps may become more vulnerable to WSSV infection because shrimps can be infected through horizontal transmission of the virus from the crabs [54]. Thus, it is very important to study regularly whether these shrimps are becoming diseased from horizontal gene transfer from carrier crabs to control WSSV infections from shrimp ghers and eradicating the disease sources. In a previous study, Hossain et al. reported higher prevalence of WSSV infections in the shrimp ghers of the Satkhira district of Bangladesh [20]. The prevalence of WSSV was 23% in the wild captured crustaceans from the south-west and south-east coast of India which included *Scylla serrata*, *Squilla mantis*, *Penaeus indicus*, and *Metapenaeus* spp. [55]. The physicochemical parameter analysis revealed that *P. monodon* can survive in a wide range of salinity (13–21 ppt), pH (7.8–9.1), dissolved oxygen (2.61–6.06 ppm), temperature (28.7–34 °C). Therefore, temperature and salinity were found to be changed very little, both in WSSV-positive and WSSV-negative shrimp ghers. In a previous study, it was found that the acceptable range of salinity, pH, dissolved oxygen and temperature can be 15–25 ppt, pH 7.5–8.5, dissolved oxygen more than 4 ppm and temperature 28–32 °C, corroborating our results [6]. We found a significant association between increase in temperature and decrease in salinity with the presence of WSSV in shrimp ghers of both Cox’s Bazar and Satkhira districts. In this study, dissolved oxygen values also had a significant association in the prevalence of WSSV. Frequent fluctuations in physicochemical factors such as pH, temperature, and dissolved oxygen make shrimps susceptible to stress, which ultimately can lead to disease [56,57]. Several lines of evidence suggested that temperature and salinity play a very important role in WSSV infection affecting the immune response of the crustaceans [57,58].

In the current study, shrimp and crab samples had huge viral loads in tissue. Remarkably, WSSV Log load per gram of shrimp and crab tissues remained much higher in samples of Satkhira district compared to Cox’s Bazar. However, comparing the viral load counts in both shrimp and crab samples, we did not find any significant difference. The presence of such high viral load risks all the shrimps in the ghers as well as the adjacent ghers, creating a possibility for WSSV outbreak in the whole area. This study suggests that WSSV load determination is essential because a shift in temperature due to any environmental reason can lead to outbreak if there is even lightly WSSV-infected crustaceans in the water body [59]. Siddique et al. reported that if few shrimps in a gher are infected with WSSV, other shrimps might be exposed over ingestion or immersion, resulting in a rapid spread of the disease leading to production disaster [29].

Phylogenetic analysis showed that VP28 sequences of Bangladesh and India fell into the major clades (phylotype BD1 and BD2). In Bangladesh, the isolated viruses showed genetic divergence falling under two different clusters (BD1 and BD2). These different clusters consisted of WSSV samples from other countries, including India, China, South Korea and Vietnam. In this study, the VP28 isolates (MZ383195- MZ383198) sequenced in 2018 (BD2 phylogroup) showed the closest genetic relatedness with previously reported VP28 isolates of Bangladesh sequenced in 2014 [20]. Likewise, two VP28 isolates (MZ383193- MZ383194) sequenced in 2019 (BD1 phylogroup) showed the closest ancestral relation to six VP28 isolates sequenced in 2015 and 2017 from Bangladesh and two Indian isolates. Thus, from the phylogenetic tree, it can be assumed that all of the isolates sequenced in the present study (2018 and 2019) were quite closely related to the VP28 sequences reported from Bangladesh in the previous years (2011–2017), and the result correlates with the previous works [20,29].

VP28 is one of the most important structural proteins of WSSV responsible for systemic infection and found to be crucial in cell recognition, attaching and penetration into the shrimp cells [60]. The aa mutation analysis showed that majority of VP28 sequences (64%) of the WSSV reported from different geographical locations (including the seven sequences of the current study) underwent mutations at 21 positions. Several earlier studies from Bangladesh [20,29] and neighbouring countries [61,62] also reported aa variations in VP28 of WSSV, supporting our current findings. In addition, residue position 167 showed glycine instead of glutamic acid (E→G) in Bangladeshi and Egyptian VP28 sequences. In both shrimp and crab isolates from the Satkhira district of Bangladesh, the unique mutation (E→G) at position 167 that falls between two beta strands of protein are thought to be involved in receptor recognition [29,62]. The exclusive aa mutation at residue position 167 of VP28 were also reported in the isolates of Bangladesh collected in 2014, 2015 and 2018 [20,29]. VP28 fuses with the host protein (PmRab7), which is the beginning of the virus–host relationship, and the viral nucleocapsid is then transported to the nucleus of host cell where the replication of the viral genome starts [62]. Sritunyalucksana et al. first mentioned that the Rab7 protein of penaeid shrimp is involved in binding an envelope protein of WSSV known as VP28 [63]. Our prediction results using DynaMut for molecule flexibility analysis had the impression that there could be rigidification in binding interactions for VP28 of BD1, and also generated a sense of possibility of stronger binding affinity with the VP28 of BD1. Kd values help to presume that there could be higher binding affinity in Rab7-VP28 complex when shrimps are infected with the BD1 phylotype. As transgenically engineered VP28 had been used in studies to build innate immunity in shrimp for its capacity to localize on host epithelial cells and attention has been given to drug designing using molecular docking and simulation studies [64], the role of mutations in VP28 are crucial to be reflected upon.

Infection assay of shrimp PL with dilution containing 10^8^ WSSV showed 97% mortality rates at 96 h of challenge irrespective of challenge with any group. Mortality patterns of PL also showed variation when exposed to different loads when challenged with both groups of WSSV. The onset of death and lethal time 50 (LT50) in the experimental PL was found inversely proportional to the dilution stock (earlier death time with less diluted stock). The mean LT50 values for the challenged PL differed between the phylogroups (BD1 and BD2) of WSSV, and remained higher in BD2-challenged shrimp PL. The average median LT50 remained lower in the case of BD1-challenged shrimp PL. Mean Log viral copy numbers were found to differ between both groups challenged PL, staying statistically higher (6.47 per mg tissue) in BD1-treated PL. Mud crabs were found to be carriers and vectors of WSSV in different countries and used for virus infectivity experiments in studies [41,42]. In our study, crabs were infected through ingestion, and it was found that crabs infected with BD1 died earlier than the ones challenged with BD2. The viral loads in all infected samples showed higher copy numbers in BD1-challenged crabs, such as BD1-challenged PL. The differences in viral load have also been stated by other authors [65,66] and can possibly be elucidated by differences in the degree of virus replication, physiological state, and defense response of the host. Higher WSSV copies and lower Ct values in BD1-challenged samples indicated that this phylotype of WSSV might contain more virus copies at the later stage of infection. There are indications that susceptibility to WSSV may differ between life stages, species and different decapods [67,68]; however, the use of a known dose of different phylotypes of WSSV is critical to demonstrate these differences. An immersion challenge to shrimp PL with an inoculum of known virus content showed that a minimum of five logs of WSSV copies is necessary to establish disease and produced a LT50 of 52 h [68]. The ingestion method for crab infections was followed in the study of Gunasekaran et al. [42] who reported that in case of crabs, the ingestion method resulted in faster deaths than the water-borne method. Although shrimps and crabs were challenged using two different methods of infection, crabs died later in our study. This may be because the crabs are a hardy species and carry the virus for a long time, and on the other hand, shrimps die quickly within 3–10 days after infection, and it is also of utmost important to consider that we infected the post larvae of shrimps which are not so resilient to diseases as those in early stage of life. Cumulative mortality was observed in shrimp infection assays using immersion or per os inoculation to be 100% at 108 hpi with different doses and the LT50 of low to high doses were 65, 57 and 50 hpi in a study conducted by Escobedo-Bonilla et al. [10].

Gene expression profiling of penaeidin and lysozyme was performed to support the findings of infection assays of the current study. Penaeidin is an antimicrobial peptide explicitly observed in penaeid shrimp. which are commonly known to show antibacterial and antifungal actions, and was reported to perform a potential part in antiviral immunity of shrimps exposed to WSSV [69]. It was observed that in *P. monodon* PL-challenged groups, there was a differential pattern of gene expression, which suggested that transcription could be due to two stages of protection mechanism, killing of microorganisms and wound healing [70,71,72,73,74]. Penaeidin’s C-terminal cysteine-rich domain with its amphipathic shape may perform as the domain for the binding pathogen. Lysozyme is another important AMP that is involved in the host-resistance arrangement of invading microorganisms [75,76,77]. In a previous study with blue shrimp (*Litopenaeus stylirostris*), lysozyme was found to be upregulated in WSSV-infected shrimp, suggesting its involvement in the innate immune response of shrimp to WSSV [78]. In the current study, the average relative gene expressions of penaedin and lysozyme in both infected groups expressed at a low level in the BD1-challenged PL with 100% mortality were quicker at producing more virus copies.

## 5. Conclusions

The current study investigated the prevalence and virulence properties of circulating WSSV in Bangladesh. The prevalence of WSSV was found to differ significantly according to hosts (i.e., shrimp and crabs), geographic locations of the ghers (Cox’s Bazar vs. Satkhira districts), and also during the time periods (2017 to 2019). The in vivo infection assay of the shrimp PL with BD1 phylotype showed an earlier LT50 and LT100 and higher viral load compared to those challenged with BD2. The AMP, penaeidin and lysozyme expression was lower in the BD1-challenged group compared to BD2. The findings of the present study revealed that the relative virulence properties of the WSSV could vary depending on the VP28 gene-based phylotypes (BD1 > BD2). Extensive investigation on the prevalence of WSSV throughout the country recruiting a larger sample group and geoclimatic conditions could illustrate more about the occurrence of this deadly virus in the semi-intensive and improved traditional ghers of Bangladesh. Moreover, changes or fluctuations in different physicochemical parameters are crucial to be observed regularly in the ghers, which could be important factors related to WSSV infection, and to understand the vibrant biological systems of host crustaceans infected with different groups of WSSV. Although it is still early to draw a conclusion on the virulence of these phylotypes, the results of the present study would be worthwhile for taking precautions in shrimp farms against WSSV and may shed new light to mitigate the huge economic loss every year in the shrimp farming of Bangladesh.

## Figures and Tables

**Figure 1 microorganisms-10-00191-f001:**
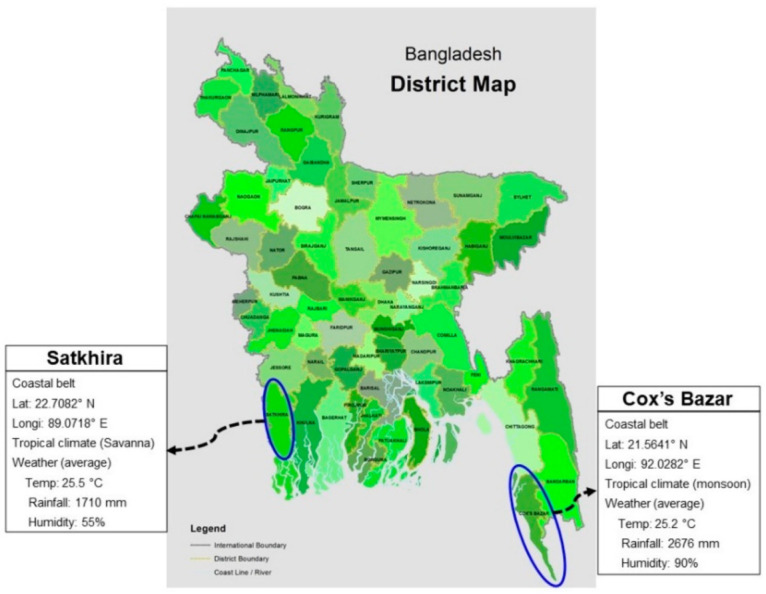
South-east and south-west coastal regions of Bangladesh (marked using blue colour) where shrimp ghers are located.

**Figure 2 microorganisms-10-00191-f002:**
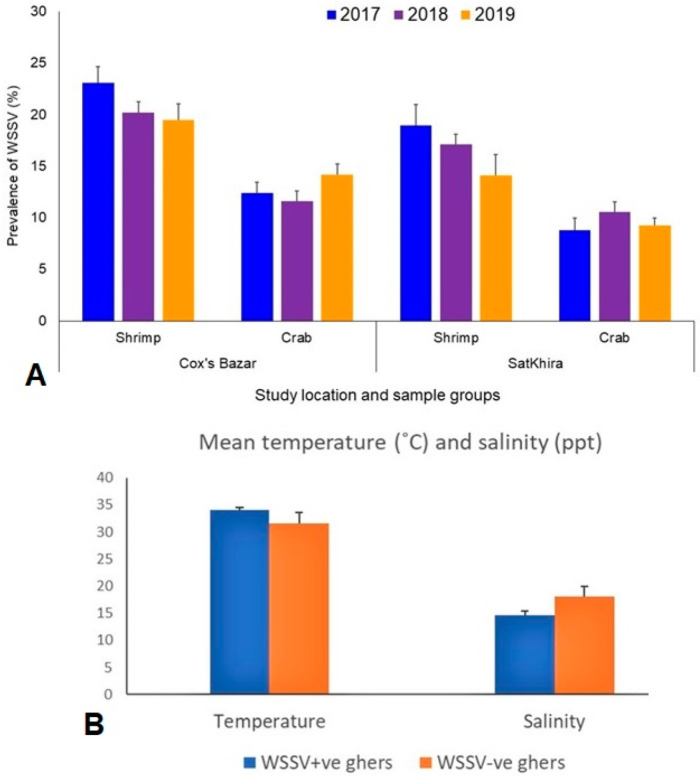
Prevalence of WSSV in Bangladesh. (**A**). The overall prevalence of WSSV in shrimp and crabs in two different regions (Cox’s Bazar and Satkhira) of Bangladesh during 2017–2019. (**B**) Mean temperature and salinity of shrimp ghers which differed significantly between WSSV+ve and WSSV-ve ghers.

**Figure 3 microorganisms-10-00191-f003:**
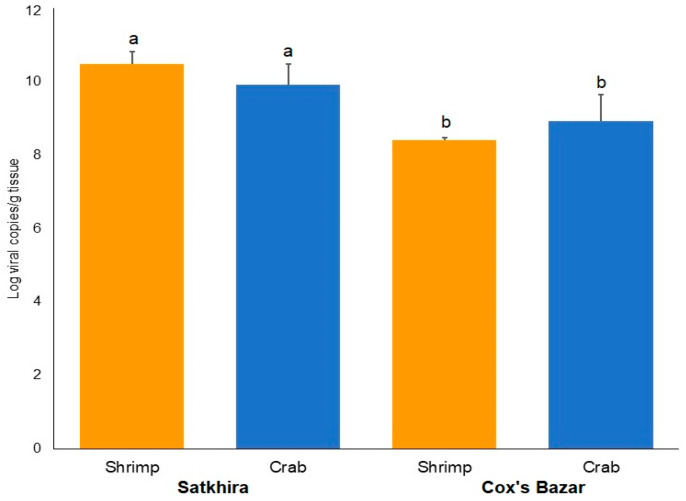
Viral load in crustacean (shrimp and crab) samples in Cox’s Bazar and Satkhira. Error bars represent standard deviation, and superscripts (a, b) represent significant differences (*p* < 0.05) statistical analysis.

**Figure 4 microorganisms-10-00191-f004:**
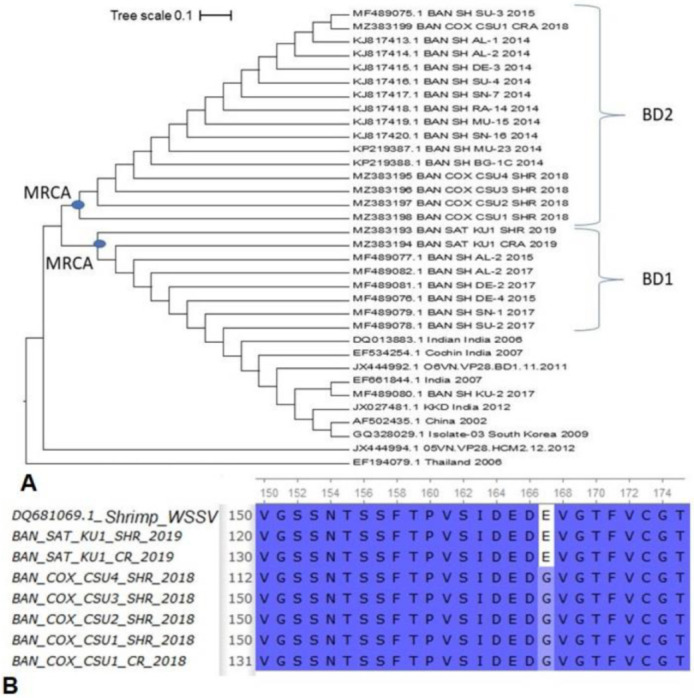
VP28 gene-based phylogenetic analysis. (**A**) Two phylotypes (BD and BD2) are currently circulating across the country. (**B**) Amino acid (aa) mutations in the VP28 sequences of WSSV.

**Figure 5 microorganisms-10-00191-f005:**
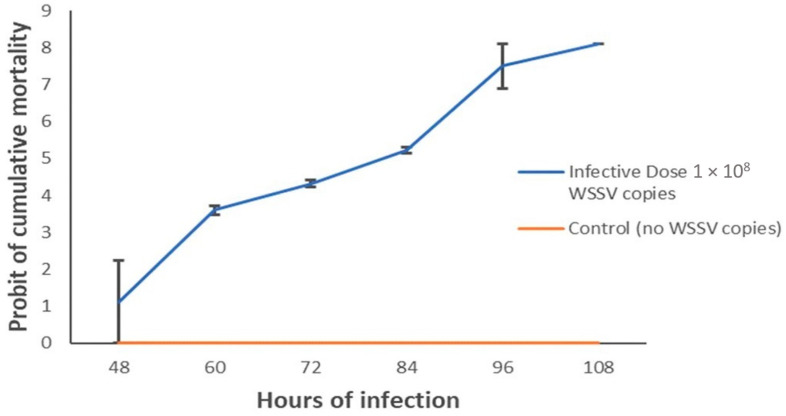
Probit of mortality. The *Y*-axis represents the morality rates while the *X*-axis shows the h of challenge.

**Figure 6 microorganisms-10-00191-f006:**
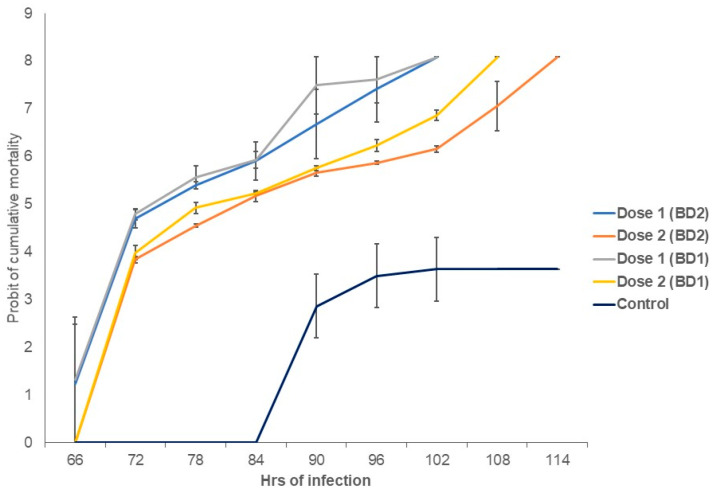
Probit of cumulative mortality after challenge with BD1 and BD2 phylotypes. The *Y*-axis represents the morality rates while the *X*-axis shows the h of challenge.

**Figure 7 microorganisms-10-00191-f007:**
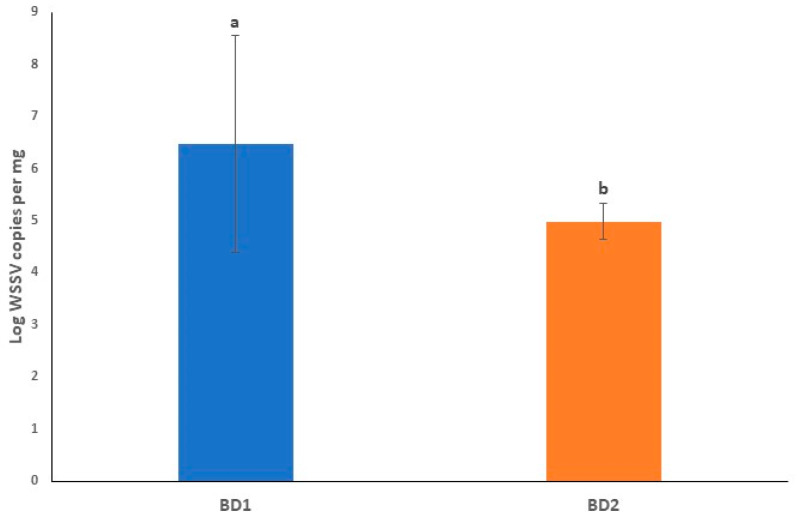
Log WSSV copies per mg of tissue infected with BD1 and BD2. Error bars and ‘alphabets’ represent the standard deviation and significant difference (*p* < 0.05), respectively.

**Figure 8 microorganisms-10-00191-f008:**
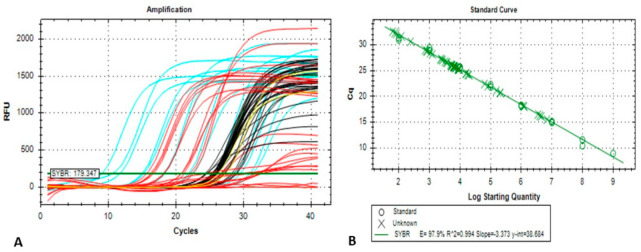
(**A**) Amplification curves targeting VP28 of WSSV and (**B**) standard curve produced using quantitative real-time PCR. Plasmid samples with known concentrations were used to obtain the standard curve, and copy numbers of unknown samples were calculated comparing Cycle Threshold (C_T_) values of samples and standards.

**Figure 9 microorganisms-10-00191-f009:**
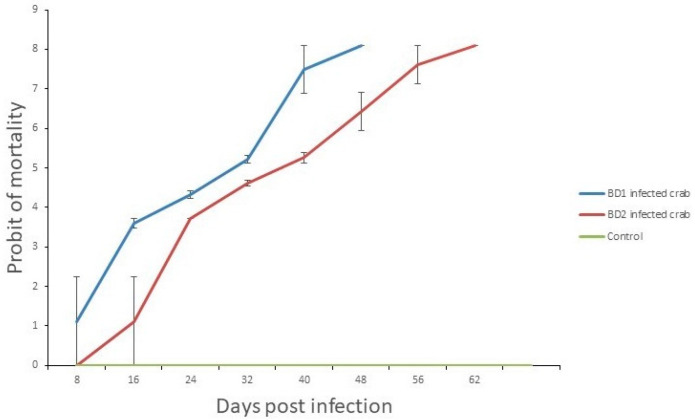
Probit of crab mortality after infection with ingestion method using a dose prepared from BD1- and BD2-infected crab tissue, and control (WSSV-negative crab tissue). The *Y*-axis represents the morality rates, while the *X*-axis shows the days of post challenge.

**Figure 10 microorganisms-10-00191-f010:**
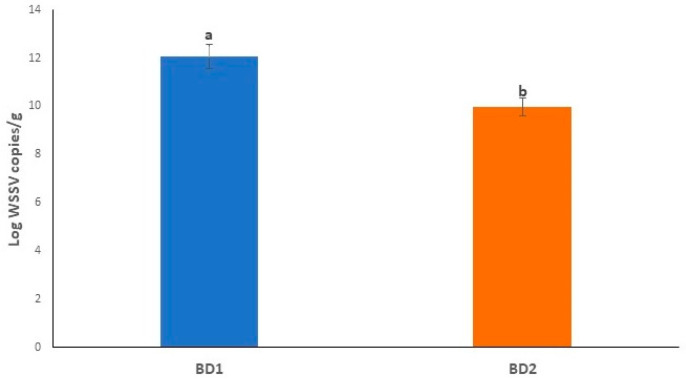
Log WSSV copies per gram tissue of infected crabs of both groups and control. Error bars and ‘alphabets’ represent standard deviation and significant difference (*p* < 0.05), respectively.

**Figure 11 microorganisms-10-00191-f011:**
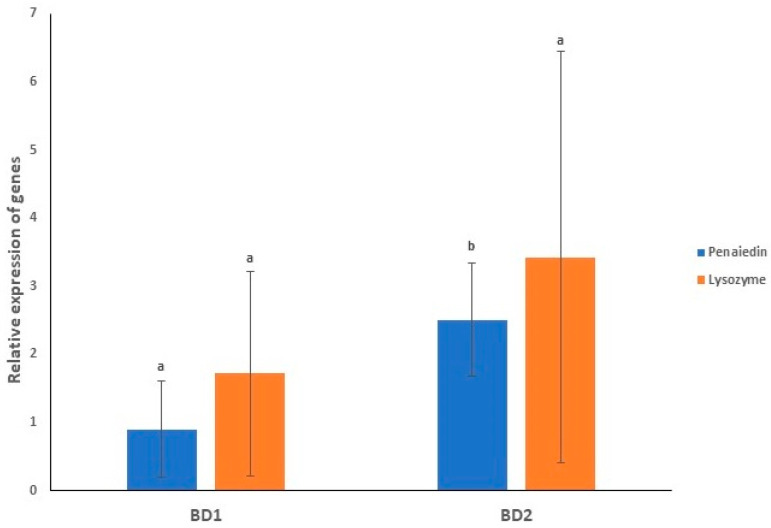
Average relative expression of immunity genes (Penaeidin and lysozyme) in shrimp PL after exposure to WSSV. Error bars and ‘alphabets’ represent standard error of means and significant differences between the expression of immunity genes, respectively.

**Table 1 microorganisms-10-00191-t001:** Primers used in the current study.

Name of Gene	Primer Sequence (5′-3′)	Reference
Penaeidin	F: TGGTCTGCCTGGTCTTCCT R: AAGCACGAGCTTGTAAGGG	[44]
Lysozyme	F: TGGTGTGGCAGCGATTATG R: GATCGAGGTCGCGATTCTTAC	[44]
Beta-actin	F: CCCTGTTCCAGCCCTCATT R: GGATGTCCACGTCGCACTT	[45]
VP28	F: GCGCGCGGATCCAATCATGGATCTTTCTTTCAC R: GCGCGCGAATTCTTACTCGGTCTCAGTGCC	[30]
qVP28	F: TGTGACCAAGACCATCGAAA R: CTTGATTTTGCCCAAGGTGT	[29]

**Table 2 microorganisms-10-00191-t002:** Physicochemical parameters in the shrimp and crab ghers.

Gher ID	Salinity (ppt)	Dissolved Oxygen (ppm)	Temperature (°C)	pH	Shrimp Samples Selected	Crab Samples Selected
Cox1	14	3.8	33.9	8.4	18	6
Cox2	21	3.75	33.7	8	18	6
Cox3	21	3.65	33.7	7.8	18	6
Cox4	20	3.90	33	8.1	18	6
Cox5	20	3.70	32.9	8.2	18	6
SS1	14	3.8	34	8.5	18	6
SS2	15	3.6	33.7	8.6	18	6
SS3	15	3.7	34	7.8	18	6
D1	19	3.8	33.8	7.8	18	6
D2	17.5	3.7	33.5	8	18	6
D3	20	3.6	33	8.6	18	6
A1	16.5	3.5	33.3	7.7	18	6
A2	16	3.8	33.8	7.8	18	6
A3	15	3.6	34	8	18	6
K1	14	3.8	34.6	8.6	18	6
K2	15	3.6	34.7	7.8	18	6
K3	16	3.6	33.9	7.9	18	6
S1	13	3.6	33.9	7.9	18	6
S2	16	3.8	34	8.1	18	6
S3	16	3.7	34	8	18	6

Gher prefix started with C represents the ghers from Cox’s Bazar while others represent the ghers of Satkhira districts.

**Table 3 microorganisms-10-00191-t003:** C_T_ values and corresponding WSSV copies from the quantitative real-time PCR (E and G of sample IDs mean samples challenged with BD1 and BD2).

Sample ID	CT	CT	CT Mean	CT SD	WSSV Copies	WSSV Copies	Mean WSSV Copies/Reaction	Mean WSSV Copies/mg Tissue
G1	27.09	27.21	27.15	0.085	2.74 × 10^3^	2.52 × 10^3^	2.63 × 10^3^	2.59 × 10^4^
G2	25.61	25.9	25.755	0.205	7.53 × 10^3^	6.21 × 10^3^	6.87 × 10^3^	8.41 × 10^4^
G3	28.80	29.02	28.91	0.156	8.50 × 10^2^	7.35 × 10^2^	7.93 × 10^2^	1.03 × 10^4^
G4	25.83	25.77	25.80	0.042	6.45 × 10^3^	6.74 × 10^3^	6.60 × 10^3^	7.76 × 10^4^
G5	25.29	25.38	25.34	0.064	9.37 × 10^3^	8.78 × 10^3^	9.08 × 10^3^	5.56 × 10^4^
G6	26.05	25.77	25.91	0.198	5.55 × 10^3^	6.72 × 10^3^	6.14 × 10^3^	6.94 × 10^4^
G7	25.30	25.35	25.325	0.035	9.31 × 10^3^	8.98 × 10^3^	9.15 × 10^3^	1.28 × 10^5^
G8	26.10	26.67	26.385	0.403	5.37 × 10^3^	3.64 × 10^3^	4.51 × 10^3^	4.16 × 10^4^
G9	28.21	28.47	28.34	0.184	1.27 × 10^3^	1.07 × 10^3^	1.17 × 10^3^	1.30 × 10^4^
G10	25.89	26.05	25.97	0.113	6.20 × 10^3^	5.58 × 10^3^	5.89 × 10^3^	6.93 × 10^4^
G11	25.74	24.49	25.12	0.884	6.87 × 10^3^	1.61 × 10^4^	1.15 × 10^3^	6.89 × 10^4^
G12	25.85	26.09	25.97	0.17	6.36 × 10^3^	5.40 × 10^3^	5.88 × 10^3^	7.20 × 10^4^
G13	25.64	25.86	25.75	0.156	7.36 × 10^3^	6.34 × 10^3^	6.85 × 10^3^	7.34 × 10^4^
G14	25.32	25.36	25.34	0.028	9.16 × 10^3^	8.89 × 10^3^	9.03 × 10^3^	6.85 × 10^4^
G15	24.30	24.52	24.41	0.156	1.83 × 10^4^	1.58 × 10^4^	1.71 × 10^4^	2.44 × 10^5^
G16	25.70	25.46	25.58	0.170	7.07 × 10^3^	8.33 × 10^3^	7.67 × 10^3^	7.55 × 10^4^
G17	24.52	24.70	24.61	0.127	1.58 × 10^4^	1.40 × 10^4^	1.49 × 10^4^	1.82 × 10^5^
G18	27.10	26.92	27.01	0.127	2.72 × 10^3^	3.07 × 10^3^	2.89 × 10^3^	3.77 × 10^4^
G19	27.40	27.04	27.22	0.255	2.22 × 10^3^	2.83 × 10^3^	2.5 × 10^3^	2.95 × 10^4^
G20	24.90	24.68	24.79	0.156	1.22 × 10^4^	1.42 × 10^4^	1.32 × 10^4^	8.06 × 10^4^
G21	24.27	24.37	24.32	0.071	1.88 × 10^4^	1.75 × 10^4^	1.81 × 10^4^	2.05 × 10^5^
G22	23.98	23.48	23.73	0.354	2.29 × 10^4^	3.22 × 10^4^	2.71 × 10^4^	3.79 × 10^5^
G23	24.00	24.26	24.13	0.184	2.26 × 10^4^	1.89 × 10^4^	2.06 × 10^4^	1.91 × 10^5^
G24	26.53	27.03	26.78	0.354	4.01 × 10^3^	2.85 × 10^3^	3.38 × 10^3^	3.76 × 10^4^
G25	23.85	24.35	24.1	0.354	2.50 × 10^4^	1.78 × 10^4^	2.10 × 10^4^	2.48 × 10^5^
G26	20.95	21.23	21.09	0.198	1.81 × 10^5^	1.49 × 10^5^	1.64 × 10^5^	9.87 × 10^5^
G27	24.63	25.09	24.86	0.325	1.47 × 10^4^	1.07 × 10^4^	1.25 × 10^4^	1.54 × 10^5^
G28	23.50	23.88	23.69	0.269	3.17 × 10^4^	2.45 × 10^4^	2.79 × 10^4^	2.99 × 10^5^
G29	23.60	24.08	23.84	0.339	2.96 × 10^4^	2.14 × 10^4^	2.52 × 10^4^	1.91 × 10^5^
G30	22.22	22.62	22.42	0.283	7.61 × 10^4^	5.79 × 10^4^	6.63 × 10^4^	9.48 × 10^5^
E1	33.48	23.65	28.565	6.951	3.48 × 10^1^	4.24 × 10^4^	2.12 × 10^4^	9.50 × 10^4^
E2	32.19	32.35	32.27	0.113	8.44 × 10^1^	7.54 × 10^1^	7.99 × 10^1^	8.88 × 10^2^
E3	31.15	29.51	30.33	1.16	1.71 × 10^2^	5.24 × 10^2^	3.48 × 10^2^	1.86 × 10^3^
E4	12.37	7.65	10.01	3.338	6.0 × 10^7^	1.59 × 10^9^	8.25 × 10^8^	1.10 × 10^10^
E5	18.23	18.11	18.17	0.085	1.16 × 10^6^	1.25 × 10^6^	1.21 × 10^6^	9.04 × 10^6^
E6	30.63	24.20	27.415	4.547	2.44 × 10^2^	1.97 × 10^4^	9.97 × 10^3^	1.03 × 10^5^
E7	7.19	28.84	18.015	15.31	2.17 × 10^9^	8.29 × 10^2^	1.09 × 10^9^	7.15 × 10^9^
E8	16.38	16.58	16.48	0.141	4.10 × 10^6^	3.56 × 10^6^	3.83 × 10^6^	2.74 × 10^7^
E9	26.82	26.59	26.705	0.163	3.30 × 10^3^	3.86 × 10^3^	3.58 × 10^3^	3.07 × 10^3^
E10	23.36	23.65	23.505	0.205	3.48 × 10^4^	2.85 × 10^4^	3.17 × 10^4^	1.62 × 10^5^
E11	20.76	20.74	20.73	0.014	2.06 × 10^5^	2.08 × 10^5^	2.07 × 10^5^	5.18 × 10^5^
E12	22.48	22.08	22.28	0.283	6.37 × 10^4^	8.33 × 10^4^	7.35 × 10^4^	1.91 × 10^5^
E13	16.22	15.60	15.91	0.438	4.57 × 10^6^	6.99 × 10^6^	5.78 × 10^6^	4.08 × 10^7^
E14	16.92	21.17	19.05	3.005	3.38 × 10^7^	1.56 × 10^5^	1.70 × 10^7^	7.78 × 10^7^
E15	32.60	31.65	32.125	0.672	6.35 × 10^1^	1.21 × 10^2^	9.23 × 10^1^	3.31 × 10^2^
E16	23.96	23.34	23.65	0.438	2.32 × 10^4^	3.54 × 10^4^	2.87 × 10^4^	1.28 × 10^5^
E17	12.21	12.53	12.27	0.226	7.06 × 10^7^	5.67 × 10^7^	6.78 × 10^7^	7.53 × 10^8^
E18	23.23	23.73	23.48	0.354	3.82 × 10^4^	2.71 × 10^4^	3.22 × 10^4^	1.72 × 10^5^
E19	12.05	12.69	12.37	0.453	7.87 × 10^7^	5.09 × 10^7^	6.33 × 10^7^	8.44 × 10^8^
E20	15.44	15.20	15.32	0.170	7.78 × 10^6^	9.17 × 10^6^	8.45 × 10^6^	6.34 × 10^7^
E21	12.00	11.26	11.63	0.523	8.15 × 10^7^	1.35 × 10^8^	1.05 × 10^8^	1.09 × 10^9^
E22	19.88	19.38	19.63	0.354	3.75 × 10^5^	5.29 × 10^5^	4.46 × 10^5^	2.94 × 10^6^
E23	12.02	12.42	12.22	0.283	8.04 × 10^7^	6.12 × 10^7^	7.01 × 10^7^	5.01 × 10^8^
E24	21.95	22.39	22.17	0.311	9.14 × 10^4^	6.77 × 10^4^	7.87 × 10^4^	6.75 × 10^4^
E25	26.40	26.82	26.61	0.297	4.38 × 10^3^	3.29 × 10^3^	3.80 × 10^3^	1.95 × 10^4^
E26	16.02	16.86	16.44	0.594	5.24 × 10^6^	2.95 × 10^6^	3.93 × 10^6^	9.83 × 10^6^
E27	12.25	11.79	12.02	0.325	6.87 × 10^7^	9.40 × 10^7^	8.04 × 10^7^	2.09 × 10^8^
E28	18.40	16.44	17.42	1.386	1.03 × 10^6^	3.93 × 10^6^	2.01 × 10^6^	1.42 × 10^7^
E29	17.10	16.74	16.92	0.255	2.51 × 10^6^	3.20 × 10^6^	2.83 × 10^6^	1.30 × 10^7^
E30	16.70	16.24	16.47	0.325	3.29 × 10^6^	4.51 × 10^6^	3.85 × 10^6^	1.38 × 10^7^

## Data Availability

The reported WSSV sequences of this study have been submitted to the GenBank database under the accession numbers MZ383193 to MZ383199.

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
