# Peer review of "Circulating Phylotypes of White Spot Syndrome Virus in Bangladesh and Their Virulence"

_microorganisms, 2022, doi:10.3390/microorganisms10010191_

Round 1
Reviewer 1 Report
The manuscript describes WSSV infection in shrimp which is an important disease contributing to substantial economic losses. Although the manuscript contains very interesting research, which is described in detail, I have some questions and suggestions.
The introduction is well written but there are some minor language errors for example “…higher disease resistant capacity” should be changed to “higher disease resistance capacity” or “infection of the WSSV on shrimp…” should be changed to “…in shrimp…”. The symbol of “°C” differs throughout the text and it should be unified. There are also some issues with citations in the manuscript – numerical citations or whole names with year citations. In chapter 2.2 about DNA extraction, there is the term “challenged shrimp” but if I understand correctly the authors means naturally infected shrimp collected from farms which is confusing. There is a lot of information regarding the parameters of water, but I would like to know at what time of year the samples were taken in field studies. The same number of samples was taken from each farm but there is no information concerning the total number of shrimp/crab in the given facility. Does “Cox 1, 2, 3, 4, 5” indicate the same farm but different times of sampling? If so why the samples from this farm were taken five times, compared to 3 times in other farms? Does all of the collected animals had clinical symptoms of the disease? In the experimental challenge, there is information concerning the origin and health status of shrimp but there is little information regarding the crab challenge. How old were the animals? Where were they tested before the challenge? Also, the number of crabs in the experimental challenge is only 5 animals, which is much less compared to the shrimp challenge. I don’t understand what “vs” stands for in figure 1B. Can you explain it?
Author Response
Reviewer’s Comment: The introduction is well written but there are some minor language errors. Our Response: We would like to thank the Reviewer for complementing our study, design, and writing, as well as the suggestion for some modifications. The language errors in the Introduction part have been corrected accordingly. You may kindly go through the Introduction section of the revised manuscript where necessary changes are highlighted in Red.
Reviewer’s Comment: The symbol of “°C” differs throughout the text and it should be unified.
Our Response: We do sincerely appreciate the reviewer comment. ËšC has been written by putting a space before it and have been unified in the whole manuscript. Please go through the revised manuscript.
Reviewer’s Comment: There are also some issues with citations in the manuscript – numerical citations or whole names with year citations.
Our Response: We would like to thank the Reviewer for pointing out this inconsistency. We have edited the citations issue in the revised manuscript keeping uniform style. You are requested to go through the revised manuscript.
Reviewer’s Comment: In chapter 2.2 about DNA extraction, there is the term “challenged shrimp” but if I understand correctly the authors means naturally infected shrimp collected from farms which is confusing.
Our Response: We are happy to the Reviewer for this judicious query. We collected samples from naturally infected and artificially challenged shrimp PL. In the chapter 2.2 about DNA extraction, the term ‘challenged shrimp’ has been used to mean the DNA extraction from experimentally challenged shrimp PL. We have revised this sub-section with addition of a sentence. You may kindly go through Page 4 in the revised manuscript.
Reviewer’s Comment: There is a lot of information regarding the parameters of water, but I would like to know at what time of year the samples were taken in field studies. The same number of samples was taken from each farm but there is no information concerning the total number of shrimp/crab in the given facility.
Our Response: We thank the Reviewer for this nice suggestion. A total of 360 shrimps and 120 crabs were grossly collected after farmers complain regarding death of the crustaceans in their farming ghers (shrimp ponds) during monsoon (May-June) in 2017 to 2019. We also continued our sampling in postmonsoon (October) during three-year study period (2017 -2019) when no death of crustaceans was reported in the study ghers by the respective farmers (Please see Table S1). The physicochemical parameters mentioned in the Tables are mean values from different years. Furthermore, the total number of shrimps and crabs in the given facility was unknown for the farmers as we collected samples from improved traditional ghers.
Reviewer’s Comment: Does “Cox 1, 2, 3, 4, 5” indicate the same farm but different times of sampling? If so why the samples from this farm were taken five times, compared to 3 times in other farms?
Our Response: We do appreciate the Reviewer for such a noble query. ‘Cox1, 2, 3, 4, 5’ does not indicate the same farm. These are the shrimp ghers of Cox’s Bazar sadar Upazilla from Cox’s Bazar district.
Reviewer’s Comment: Does all of the collected animals had clinical symptoms of the disease? In the experimental challenge, there is information concerning the origin and health status of shrimp but there is little information regarding the crab challenge.
Our Response: Thank you very much for this query. All the collected samples did not have clinical symptoms of the disease at the same time. Moreover, farmers could not confirm the health status of the shrimps and crabs in the given facility. For the crab challenge we needed WSSV-free crabs which were collected from WSSV-free crab farms in Satkhira and they were also further tested for WSSV-negativity confirmation using conventional PCR.
Reviewer’s Comment: How old were the animals? Where were they tested before the challenge?
Our Response: We thank the Reviewer for this nice comment. The crabs were of 6 months of age. Crabs those were used for the in-vivo challenge tests were collected from WSSV-free crab farms. Their WSSV-negativity was further tested before the experimental infection using conventional PCR in the lab. You may kindly go through the revised manuscript.
Reviewer’s Comment: Also, the number of crabs in the experimental challenge is only 5 animals, which is much less compared to the shrimp challenge.
Our Response: We are happy to see the reviewer’s comment. It has been a pilot study conducted on 45 mud crabs using the tissues from both phylotypes (BD1 and BD2) through ingestion method. The crabs were collected from a WSSV-free crab farm of Satkhira district. Moreover, we also tested the health status of the crabs through conventional PCR before the experimental infection. The crabs were divided into three groups; Group-1 = 15 crabs challenged with BD1, Group-II = 15 crabs challenged with BD2, and Group-III = 15 crabs, served as control. The crabs were of 6 months of age, and each group was kept in separate tank containing 5 crabs in each. You may kindly go through the revised manuscript.
Reviewer’s Comment: I don’t understand what “vs” stands for in figure 1B. Can you explain it?
Our Response: Thank you for this nice comment. The word ‘vs’ was used to compare between two groups which is not actually necessary in the figure and has been deleted. Please go through the revised manuscript.

Reviewer 2 Report
The microorganisms_1506483 manuscript analyzes the White Spot Syndrome Virus philotypes circulating in 2 areas of Bangladesh and evaluates their virulence on two species, Black tiger shrimp Penaeus monodon and mud crab Shylla olivacea, by means of experimental tests.
The topic is of great interest as it is always useful to map the presence of pathogenic viruses in all the countries in which they occur in order to outline the distribution in a detailed manner; moreover, this work goes into detail by identifying the philotypes present in the area. Although it is a work that describes a local situation, it is always very important to have an overview of the health issues affecting aquaculture in order to broaden general knowledge.
Given that the paper is extremely interesting and the idea of evaluating the virulence on the two species reared in the territory under consideration by means of an experimental test is a winning one for giving more body to the work, I have noticed some critical issues that need to be corrected.
The introduction is well done and also takes into consideration the various general characteristics of the production of these 2 crustaceans. On page 3 on the eleventh line, the citation given must be changed to Siddique and coll. (29), to standardize the method of entering bibliographic citations.
In the chapter on M&M, the two areas in which the researches were conducted are first briefly described. It is necessary to expand the description of the different areas to better characterize the sampling sites; in the second line of the sub-chapter insert “upazilla” correctly; furthermore, the figure indicated in the supplementary material (figure S1) should in my opinion be inserted directly into the text in order to make the geographical area easier to understand; however, the inserted figure must have a clearer view than the inserted one. As regards the sample size it is sufficiently large (360 shrimps and 120 crabs), but considering that the monitoring time is between 2017 and 2019 (3 years?) and that the monitored “ghers” were 20, precisely because of the importance of health monitoring, the authors should have made a greater sampling effort also to give greater robustness to the data obtained.
In the sub-chapter of the experimental infection, the description of the protocol used is not always clear: it is necessary to insert in order all the notions to be reported in a clear and easy to understand way. First, it should be noted where the individuals used come from and the total number used, how many tanks were used for the test (considering the tests carried out, the negative control and any replicas) with the environmental parameters used. Then everything else must be described. From what can be deduced, the sample size appears very low and barely sufficient to give a positive evaluation to the test. On page 6, fifth line, the bibliographic citations must be standardized by inserting the corresponding number (42). In sub-chapter 2.8, all bibliographic citations must be formatted as required by the editing with the corresponding number, but some have not been included in the list of references (Vangone & Bonvin, 2015; Xue et al., 2016; van Zundert et al., 2016 ; Biasini et al., 2014; Rodrigues et al., 2018). On the eighth line, instead, the number 55 must be inserted in correspondence with Verma et al., 2013.
The results chapter is well detailed and described and sufficiently clear.
The discussion is also well articulated and sufficiently clear, but there are numerous oversights on the part of the authors that need to be remedied. On page 18 forty-fifth line, remove the date in brackets. On the next two pages, the bibliographic citations inserted must be formatted as necessary: on page 19, the twenty-sixth line, the numbers 20 (Hossain et al., 2015) and 29 (Siddique et al., 2018) must be inserted; on the thirtieth line insert the number 55 (Verma et al., 2013) and add the bibliographic citation (Sritunyalucksana et al., 2006) to the references; on the forty-first line, add the bibliographic citation Chandrika & Puthiyedathu, 2021 to the references. On page 20, third line insert numbers 41 (Chen et al., 2000) and 42 (Gunasekaran et al., 2018) in place of the citations described. On the twentieth line insert the number 42 in brackets instead of the date, as well as on the thirty-first line the number 10. On the thirty-seventh line, add the bibliographic citation Woramongkolchai et al., 2011, as well as on lines 41-42 (Kawabata et al., 1996; Bachere et al., 2000; Munoz et al. 2002; Li et al., 2010; Song & Li, 2014), 46-47 (Sotelo-Mundo et al., 2003; Xing et al., 2009; Liu et al., 2017) and 50 (Mai & Wang, 2010).
Please note that the inclusion of new bibliographic citations will give a new numbering to subsequent citations which must be correctly reported in the text. I ask for greater attention to the authors for this operation. A final note for the authors: among the supplementary material you have inserted the table of the primers used (table S3); for greater clarity and completeness, I would insert this table in the text indicating the references with the number relating to the citation: even the missing ones to be included in the final chapter of the references (Deris et al., 2020 and Shekhar et al., 2015).
For all these reasons, in my opinion, despite the goodness of the work, major revision is required to remedy all the shortcomings noted.
Round 2
Reviewer 2 Report
The authors of the paper Microorganisms-1506483 have satisfied the requests of the referees obtaining a more than satisfactory result.
Depsite the revisions made, I could still notice a small error to correct that does not affect the value of the text and does not require, in my opinion, a further round of revision; on page 20, third to last line, the bibliographic references must be placed in increasing numerical order: 20, 29 and not vice versa.
For these reasons, the manuscript can be accepted for publication in Microorganisms.
Author Response
Reviewer’s Comment: Depsite the revisions made, I could still notice a small error to correct that does not affect the value of the text and does not require, in my opinion, a further round of revision; on page 20, third to last line, the bibliographic references must be placed in increasing numerical order: 20, 29 and not vice versa.
Our Response: We would like to thank the Reviewer for complementing our study, design, and writing, as well as the suggestion for modifications The bibliographic references have been corrected accordingly. You may kindly go through page 20 (highlighted in red) in the revised manuscript.
